# Auto-Tiler: Variable-Dimension Autoencoder with Tiling for Compressing Intermediate Feature Space of Deep Neural Networks for Internet of Things

**DOI:** 10.3390/s21030896

**Published:** 2021-01-29

**Authors:** Jeongsoo Park, Jungrae Kim, Jong Hwan Ko

**Affiliations:** 1Department of Electronic and Electrical Engineering, Sungkyunkwan University, Suwon 16419, Gyeonggi-do, Korea; jspark702@gmail.com; 2Department of Semiconductor Systems Engineering, Sungkyunkwan University, Suwon 16419, Gyeonggi-do, Korea; dale40@skku.edu

**Keywords:** collaborative intelligence, deep feature compression, inference partitioning, autoencoder, convolutional neural network, deep learning, machine learning, distributed computation, Internet of Things

## Abstract

Due to limited resources of the Internet of Things (IoT) edge devices, deep neural network (DNN) inference requires collaboration with cloud server platforms, where DNN inference is partitioned and offloaded to high-performance servers to reduce end-to-end latency. As data-intensive intermediate feature space at the partitioned layer should be transmitted to the servers, efficient compression of the feature space is imperative for high-throughput inference. However, the feature space at deeper layers has different characteristics than natural images, limiting the compression performance by conventional preprocessing and encoding techniques. To tackle this limitation, we introduce a new method for compressing DNN intermediate feature space using a specialized autoencoder, called *auto-tiler*. The proposed auto-tiler is designed to include the tiling process and provide multiple input/output dimensions to support various partitioned layers and compression ratios. The results show that auto-tiler achieves 18% to 67% higher percent point accuracy compared to the existing methods at the same bitrate while reducing the process latency by 73% to 81%. The dimension variability of an auto-tiler also reduces the storage overhead by 62% with negligible accuracy loss.

## 1. Introduction

Artificial Intelligence (AI) is rapidly spreading into Internet of Things (IoT) devices, including face recognition for smart security systems [1,2,3], voice assistant with AI speakers [4,5,6], and smart cars [7,8]. IoT edge devices, however, do not have sufficient resources to perform inference of complex deep neural networks (DNN) in a timely manner. To satisfy the latency requirement, the inference computation load can be offloaded to high-performance servers in the cloud. This offloading approach started as a full-offloading, and has evolved to a partial offloading. In full offloading, a mobile device transmits input data to a cloud server, which performs inference of the entire DNN and returns a result back to the device. The server, equipped with cutting-edge GPUs and AI accelerators is generally orders of magnitude faster than the mobile device and can meet latency and throughput requirements of real-time inferences.

As IoT edge devices enhance their AI processing capabilities, the offloading mechanism has evolved to partial offloading, called ‘Collaborative Intelligence’ [9,10,11,12]. Recent mobile devices have AI hardware units inside (e.g., neural processing unit in an application processor) and can perform inference of simple DNNs on-device. However, these units cannot execute complex DNN models by themselves due to limited resources from battery-supplied power and small form factors. Instead, collaborative intelligence utilizes the on-device hardware to keep some computation on-device, and partially offloads the rest of the computation to a server. The motivation behind collaborative intelligence is that most DNNs have intermediate feature spaces smaller than the input feature space. Collaborative intelligence utilizes this characteristic to trade computation time for communication time. It partitions DNN layers and the edge device executes up to the partitioned layer. The output features from the partitioned layer are then transferred to a cloud server, which will continue executing the rest of the network. Executing early layers on the relatively-slow edge device and transferring a smaller amount of features can reduce the end-to-end latency significantly than transferring a large volume of input data and executing the entire layers in the relatively-fast server hardware.

An offloading scheme, either full offloading or collaborative intelligence, incurs significant overheads in transferring a large volume of feature data, especially when the input data is a high-resolution image or video. Compressing the feature data is promising for reducing the communication time. There is a rich literature on compressing input image/video, including Joint Photographic Experts Group (JPEG) and High Efficiency Video Coding (HEVC). These codecs have evolved over decades and can achieve a very high compression ratio on vision data. Compressing intermediate features is relatively new and many studies [13,14,15,16] extend existing image/video codecs to compress vision-based intermediate features. They add a preprocessing stage before applying an image/video codec to fit intermediate features into the target codec.

Such studies, however, are sub-optimal by poorly processing multiple channels. In most convolutional neural networks (CNN), channel count increases and feature map size decreases as layers deepen. Some studies [13,14] individually apply a codec to these many small-sized maps, which suffers from limited redundancy and unamortized header costs. The others [15,16] tile multiple maps to build a large frame, which introduces blockiness in the combined frame and degrades the efficiency of natural-image-based codecs.

To address these limitations, this paper presents a new preprocessing technique for intermediate compression, called ‘auto-tiler’ (Figure 1). Auto-tiler is a specially-designed autoencoder that encodes an intermediate feature space (a collection of feature maps) into a single output feature map. Image/video codecs can then be applied to this map without any additional preprocessing. The output map has a smaller dimension than its inputs, and auto-tiler supports multiple output dimensions from a single model to support multiple compression ratios.

Another strength of an auto-tiler is its flexibility in dealing with changes in communication conditions. In a real environment, communication conditions (e.g., latency, throughput, and error rate) can change frequently. Such a change affects communication time and can change the optimal partition layer in collaborative intelligence. The proposed auto-tiler is designed to support multiple partition layers using a single model. The same model can be reused to process intermediate features from different partitioning layers.

Our evaluation shows that auto-tiler achieves 18% to 67% higher percent point accuracy compared to the existing methods at the same bitrate. Auto-tiler also improved the process latency by 73% to 81% depending on the compression quality. Additionally, by allowing an auto-tiler to support multiple input and output dimensions, we managed to save the storage overhead by 62% with minor accuracy loss.

The rest of the paper is organized as follows—Section 2 provides backgrounds and motivations for auto-tiler. In Section 3, we introduce the characteristics of auto-tiler and its unique design choices. In Section 4, we explain our experimentation settings and compare the performance of auto-tiler with existing methods. Section 5 provides the summary of our paper and future works.

## 2. Backgrounds and Related Work

### 2.1. Collaborative Intelligence

DNN requires billions of operations to infer. Mobile devices, which have limited power supply and computational resources, used to offload the entire computation to clouds. The cloud-only approach fully offloads the computation to the cloud and utilizes high-performance GPUs and AI accelerators to complete the job in a timely manner. As mobile devices introduced AI hardware, this offloading scheme has evolved to *Collaborative Intelligence* to partially offload computation to servers. With collaborative intelligence, server computation time is traded with on-device computation to reduce transfer time.

There are researches to compress intermediate features in collaborative intelligence in order to reduce the transfer time [13,14,15,16]. Some suggest preprocessing methods to make feature space easily compressed by conventional image/video codecs such as JPEG and HEVC. They utilize state-of-the-art codecs to achieve high compression efficiency and can be classified into two categories based on how they process multi-channel features.

### 2.2. Existing Feature Preprocessing Methods and Their Limitations

We organized the existing feature preprocessing methods into two groups—one that applies codecs individually to each feature maps, and another which first tile these feature maps into a large frame before applying codecs. In the following subsections, we will discuss these two methods in detail and analyze their limitations.

#### 2.2.1. Frame per Channel Method

Most CNNs decrease their feature map size but increase feature map count (i.e., channel count) as layers deepen. For example, input features of AlexNet [17] and VGG16 [18] have a dimension of 224 × 224 × 3 (width × height × channel). But the third convolutional layer of AlexNet has an intermediate feature space of 13 × 13 × 384 and the conv3_2 layer of VGG16 has an output dimension of 56 × 56 × 256. Some methods [13,14] regard each channel as a frame and compress separately (referred to as *Frame Per Channel (FPC)* in this paper, Figure 2). This approach becomes less efficient as the feature map size gets smaller. Figure 3 compares JPEG compression ratio of a sample image against ones of different-sized sub-images. Each sub-image is built by slicing the image into smaller sub-images. For example, the four 250 × 203 sub-images are built by splitting 500 × 406 images evenly both horizontally and vertically. The result illustrates that the compression ratio decreases as the feature map size decreases. Our evaluation in Section 4 also shows FPC-based methods require more bandwidth for a target accuracy or exhibit poor accuracy for a given bandwidth.

#### 2.2.2. Tiling Method

The others [15,16] ‘tiles’ many small-sized feature maps into a single large frame and apply a codec to the entire frame (i.e., the entire feature space) (Figure 2). With increased frame size, tiling-based methods can exploit increased redundancy and amortize costs to achieve a higher compression ratio than FPC-based ones. However, one of the drawbacks of these methods is that it introduces blockiness into the frame. At feature map boundaries, there can be abrupt changes in pixel values. The resulting frame has blocking artifacts at tile boundaries and suffer from sub-optimal compression efficiency with natural-image-based codecs, which rely on a frequency-domain transform to increase data redundancy and reduce the perception of errors [19,20].

As explained above, both existing approaches have limitations as a preprocessing method of feature map compression. Furthermore, deep feature space has different spatial characteristics compared to natural images [15]. To verify this difference in characteristics, we also conducted an experiment on a partitioned YOLO v3 network [21]. In this experiment, spatial similarity is measured as a percentage of pixels that is within 2% difference with its spatial neighbors (left, top, top-left, bottom-left, top-right). For each neighbors, the similarity is calculated individually and then averaged to determine the overall spatial similarity. As shown in Figure 4, the overall spatial similarity of the network generally decreases as the layer goes deeper. Therefore the deep feature space should be preprocessed in a way that the spatial similarity can be improved, to effectively use conventional image compression algorithms.

### 2.3. Autoencoder as a Preprocessor

Autoencoders are a special type of a neural network which is typically used to reduce the dimension of the input feature space using a bottleneck layer. They are trained to learn and approximate the identity function so that the decoding part of the network can reconstruct the input from the compressed bottleneck layer activation.

Deep feature space can be better compressed if we train and use an autoencoder network that is tailored towards compressing the intermediate features. This is because conventional compression algorithms are crafted based on realistic images with human visual system in mind, which the deep feature space differs from. Autoencoder network can then be used as a preprocessor during feature map compression by decreasing the dimension and number of channels of the feature space. This autoencoded feature space is then tiled into a single frame and further compressed by image/video encoders.

We conducted a simple experiment to check the feasibility of autoencoders as the preprocessor. Figure 4 portrays that a simple autoencoder showed relatively higher spatial similarity in deeper layers, while the spatial similarities of other preprocessing methods decreased. The increased redundancy at deeper layers will allow the video encoders to compress the feature space even further, thus improving the compression ratio.

Although autoencoder can encode the feature space while increasing the spatial similarity, the output feature space is still many small-sized channels, which are not favorable to the conventional encoders. Even though tiling can combine these channels into a single frame, it causes blockiness introduced by different feature maps being adjacent to each other in a tiled frame. The proposed method solves this problem by using an autoencoder with special bottleneck structure while preserving the advantages which autoencoder brings.

## 3. Proposed Method

We propose a new preprocessing method called ‘auto-tiler that utilizes a specially-structured autoencoder to encode the feature space into a reduced dimension. The important aspect of the proposed method is that it automatically ‘tiles’ the multi-channel feature space into a single-channel one within its encoding process. By effectively allowing an auto-tiler to learn the optimal tiling process by itself, it eliminates the additional process that was required for better compression by the existing methods. An example of an auto-tiled feature map is illustrated in Figure 5. This feature map does not have the problems arising from many smaller-sized feature maps in Figure 2b and also does not show multiple blocky edges as shown in Figure 2c. The effect of these characteristics are demonstrated in Section 4.3.1. The result indicates that the proposed method retains noticeably higher structural similarity after compression than the existing methods shown in Figure 2b,c, allowing for a more efficient compression. In addition, we design an auto-tiler to accept the input feature space with multiple different dimensions, in order to support various partitioned layers of a network. It also allows multiple output dimensions to support quality-compression scalability. The structure of a proposed auto-tiler is shown in Figure 6, and its core features (auto-tiling, variable input dimensions, and variable output dimensions) are explained in the following subsections.

### 3.1. Auto-Tiling Autoencoder

The structure of a conventional autoencoder is altered so that it outputs a single, larger feature map at the bottleneck layer. The size of a filter (convolution kernel) in the bottleneck layer is increased to compensate for the low quantity of filters. This eliminates the need to perform tiling/de-tiling operation before the encoding/decoding process. That is, the bottleneck activation of an auto-tiler can be directly encoded by a video encoder, and its decoded frame can also be directly used as an input to the decoding part of an auto-tiler. Furthermore, by using a single, larger filter, the blockiness of a tiled (in this case, auto-tiled) frame can be reduced, which makes room for a video encoder to improve the compression ratio.

### 3.2. Auto-Tiler with Variable Input/Output Dimensions

During inference, the optimal partitioned layer may change based on available network capacity and compute/memory resource of the edge device [9]. It implies that the dimension of the intermediate feature space can vary depending on the partitioned layer. However typical autoencoders can only accept the input with one fixed dimension.

Another challenge in partitioned inference is that the required compression ratio can vary depending on the latency requirement or transmission link conditions. However, the output feature dimension of conventional autoencoders is fixed, so the compression ratio cannot be altered on-line. One of the simple solutions to this issue is to train multiple autoencoder models with different input and output dimensions (compression ratios), and store all the models in an edge device. However, this is not an efficient solution as it increases storage overhead proportional to the number of required models. Another approach can be dynamically tuning an autoencoder model depending on the required input/output dimension. However, this method also is limited since training an entire autoencoder network is a costly operation. Therefore, it is advantageous to design an autoencoder (an auto-tiler in the proposed approach) so that it is compatible with variable input and output dimensions.

#### 3.2.1. Variable Input Dimensions

We design an auto-tiler with multiple input dimensions by adding compatibility autoencoders with different structures to a core auto-tiler. A core auto-tiler is a network that is trained on the intermediate layer with the smallest dimension and will be used in conjunction with compatibility autoencoders as needed. Compatibility autoencoders transform the larger input dimension to a smaller one which is compatible with a core auto-tiler. In this way, we can easily support multiple input dimensions while minimizing the storage overhead, since a large portion of the network (core auto-tiler) is shared.

We applied this approach to the YOLO v3 network, which has three partition-able output dimension (Width, Height, Channels): (208, 208, 64) for the 4th layer, (108, 108, 128) for the 11th layer, and (52, 52, 256) for 15th, 24th, and 36th layer. Partitioning beyond the 36th layer is undesirable due to the skip line concatenating the 36th layer with the 96th layer. In order to support these multiple dimensions, we have first trained a core auto-tiler network on activation feature space of 15th, 24th, and 36th layer as they have the same, smallest dimension. Then for the 4th and 11th layer, we trained additional compatibility autoencoders that reduce the dimensions so that it can be compatible with a core auto-tiler.

#### 3.2.2. Variable Output Dimensions

We designed an auto-tiler with variable output dimensions by allowing a core auto-tiler to generate different bottleneck activations in its inference process. Therefore, it will encode the input with different strengths depending on the inference depth. In this way, we can change the compression ratio without having to re-train the network.

To determine the optimal compression ratio (CR) that an auto-tiler should support, we trained a simple auto-tiler by increasing the CR by 4×, from 4× to 256×. Then, we defined an effective mAP to be 85% of the maximum mAP of the YOLO v3 network. Should an auto-tiler network with a certain CR achieves less mAP than an effective mAP, that CR will be considered ‘too heavy’. Our experiment on Figure 7 showed that a CR of 4× to 64× was within our effective mAP range, with 64× sitting on the edge of an effective mAP. Thus, we determined that the optimal CR that an auto-tiler network should support is within 4× to 64×.

Based on this approach, we designed an auto-tiler network so that it can compress the input feature space into two different output dimensions. These output dimensions are (416, 416, 1), (208, 208, 1), which compress the feature space of the 15th, 24th, and 36th layers by 4× and 16×, respectively. Its compression ratio is increased by 2× as the partition depth gets shallower. That is, when compressing the 11th layer activation the compression ratio will be 8× and 32×, and when compressing the 4th layer it would be 16× and 64× respectively. This is due to the compatibility autoencoders compressing the 11th and 4th layer activation by 2× and 4× in order to make it compatible with a core auto-tiler. As a result, an auto-tiler network will support a CR from 4× to 64×. The structure of an autoencoder, auto-tiler, and compatibility autoencoders is shown in Figure 8.

## 4. Experiment Results

### 4.1. Experiment Settings

In the following subsections, we discuss the experimentation settings. We first discuss which framework we used to evaluate and train the models. Then we describe the model structure and training setups. Subsequently, we demonstrate the configurations for the video encoder.

#### 4.1.1. DNN Model Training

The proposed autoencoders including an auto-tiler are trained using TensorFlow [22]. YOLO v3 model that we have used during training is a pre-trained network which is transfer-learned on VOC dataset [23]. As for the training dataset, we used 16,125 images from a total of 18,407 combined images of VOC2007 and VOC2012 training dataset [24,25]. We have excluded the first 2282 images in the training dataset to use during validation. Then 287 overlapping VOC2007 test images are removed and the remaining 1995 images are used as a validation dataset. Finally, for testing, we used 4952 test images from the VOC2007 test dataset. Every measurement of the proposed method are derived by averaging the results of all images in the test dataset. Additionally, all training and testing are done with PC specs of AMD Ryzen 7 3900x and NVIDIA RTX2080, which have a respective FP32 computation capability of 2649.6 GFLOPS and 10.07 TFLOPS [26,27].

We first trained a simple autoencoder that does not have an auto-tiling bottleneck layer as a comparison. The CR4 model, which gives a compression ratio of 4×, is trained initially and extra layers are then attached to further reduce the dimensions by 4× which constitutes the CR16 model as shown in Figure 8a. The encoding part of the CR16 model is trained with its parent CR4 encoder frozen. The decoding part on the other hand is trained individually for each compression ratios since server platforms are less limited by storage space than edge devices. This allows us to achieve better accuracy without sacrificing the limited storage space on edge devices. The learning rate is set to 0.003 and 0.001 when training the CR4 model and CR16 model respectively. A simple autoencoder is only trained on the 36th layer for the sake of comparison.

When training a core auto-tiler we loaded the activations from the 15th, 24th, and 36th layer sequentially since they have the same dimension. In a similar manner as training an autoencoder, a CR4 auto-tiler is trained initially. Additional layers that provide heavier compression are then attached to form a CR16 auto-tiler. The encoding part of a CR16 model is trained with its parent CR4 encoder frozen, and the decoding part is trained individually in the same vein as an autoencoder. It should be noted that this still reduces the decoders’ storage overhead since a core auto-tiler is shared and supports multiple partitioned layers through compatibility autoencoders. The learning rate was fixed to 0.0003 during training. The structure of a core auto-tiler is shown in Figure 8b.

Compatibility autoencoders for 4th (Figure 8c) and 11th (Figure 8d) layers are trained by freezing the weights of a core auto-tiler and attaching the autoencoders at both ends. We trained the CR16 models first and then trained the CR4 models afterward. When training the CR4 models, we copied the weights from the CR16 model and froze the weights of the encoding part. Therefore the CR4 and CR16 models are trained to share the encoding part while the decoding parts are trained individually to recover the original feature space. We used a learning rate of 0.0005 for the 4th layer compatibility autoencoder and 0.00001 for the 11th layer.

All of the autoencoders and auto-tiler models are trained for 60 epochs with the early stopping patience of 5. We chose a model with the lowest validation loss to use during testing.

#### 4.1.2. Video Encoder Settings

In our experiment, the feature maps created by an autoencoder or an auto-tiler are encoded by the HEVC codec, available as an FFmpeg library [28,29,30]. The frame rate was set to 30 frames per second, and the pixel format is set to grayscale. We used a bitrate based encoding where the encoding bases are [450, 900, 1800, 3600, 7200, lossless] Kbps, which are equivalent to [15, 30, 60, 120, 240, lossless] Kbpi (kilobits per image). When using the FPC method, these bitrates are divided by the number of channels to keep the Kbpi in a similar range. That is, for the FPC method on (52, 52, 256) feature space, we used the encoding bases as [450/256, 900/256, 1800/256, 3600/256, 7200/256, lossless] rounded to the nearest integer.

### 4.2. Independent vs. Variable Auto-Tiler

A variable auto-tiler model has a distinct advantage over the independent ones. That is, it reduces the storage space overhead by not requiring multiple models to support varying dimensions. However, implementing variability to a model may reduce the accuracy as a model could become less specialized to a particular configuration. Therefore in the following subsections, we illustrate that the proposed method obtains comparable accuracy even with the added variability. Then, we compare the storage space overhead of each method to highlight the advantage of a variable model.

#### 4.2.1. Compression Performance Comparison

Considering that we have designed an auto-tiler to support variable input and output dimensions, we must compare its accuracy with independent models to verify its feasibility. In Figure 9 we compared the result with independent models and achieved around 1.15% accuracy loss with our variable model. Consequently, we were able to verify the feasibility of a proposed variable model in terms of accuracy as it showed negligible accuracy loss compared to the invariable, independent models.

#### 4.2.2. Memory Overhead Comparison

The most significant advantage of a variable auto-tiler model is its much smaller storage space overhead than the independent models. We compared the memory overhead of variable and independent models in Table 1, which indicates that the overall memory overhead is reduced by 62%. The encoder part, which will be stored on edge devices, is reduced by 74%.

### 4.3. Auto-Tiler vs. Existing Methods

The following subsections will discuss several major advantages of an auto-tiler against existing methods. We first discuss the structural robustness of the proposed method to the compression artifacts. Then we compare the compression performance at various configurations and analyze the effective compression ratio of each method. Afterward, we study the latency-improving effects of the proposed method.

#### 4.3.1. Structural Similarity Comparison

To verify that our proposed method is more structurally stable after compression than the existing methods, we measured the structural similarity (SSIM) [31] of auto-tiler CR4-16, FPC and tiling method over all partitioned layers. SSIM is measured using FFmpeg library [30] with the losslessly encoded feature space as a reference data. SSIM value is within the range of −1 to 1, where 1 denotes that the target is an exact match to the reference data and −1 represents that the target is most dissimilar. The result is shown in Figure 10. It is clearly demonstrated that auto-tiler exhibits markedly higher SSIM than the existing methods over all Kbpi setting. This illustrates that our proposed method allows a more efficient compression by suffering less from compression artifacts resulting from blocky edges of tiled feature maps, or inefficient compression arising from small-sized feature maps and increased header cost of FPC method.

#### 4.3.2. Compression Performance Comparison

Comparison of compression performance between an autoencoder, auto-tiler, and the existing methods is shown in Figure 11. Auto-tiler generally showed better rate-distortion characteristics compared to Tiling and FPC method. It also suffered notably less accuracy loss from compression compared to those of autoencoder while achieving similar maximum accuracy. This is due to the fact that an auto-tiler removes the need to manually ‘tile’ the feature maps, therefore reducing blockiness and discontinuity resulting from tiling.

We also compared mAP at 60 Kbpi, Kbpi at lossless video encoding mode, mAP at lossless video encoding mode, and Kbpi required to achieve 0.6 mAP. The result is shown in Figure 12. At 60 Kbpi, the auto-tiling method showed substantially higher mAP compared to the FPC or Tiling method which only uses video encoders to compress. Versus FPC method, an auto-tiler at CR4 mode attained 23.8% to 62.1% higher percent point accuracy, and at CR16 mode it showed 53.0% to 67.3% higher percent point accuracy. However, comparing an auto-tiler results with the FPC method was nearly pointless as the accuracy of that method was near 0%. We then compared an auto-tiler with the Tiling method and achieved 18.2% to 46.1% higher percent point accuracy when using CR4 mode, and 30.8% to 57.3% higher percent point accuracy when using CR16 mode. When losslessly encoded using HEVC codec, CR4 quality auto-tiler achieved 3.2× to 9.0× higher compression ratio compared to those of Tiling method, and CR16 quality auto-tiler achieved 22.0× to 32.8× higher compression ratio. When compared to the FPC method, CR4 mode auto-tiler achieved 6.1× to 11.6× higher compression ratio, and CR16 mode achieved 11.4× to 42.3× higher compression ratio.

The accuracy of an auto-tiler is reduced due to being a lossy compression method. A CR4 quality auto-tiler suffered an average of 2.3% mAP loss where 4th layer accuracy loss was the highest as a result of having the highest compression ratio. An average mAP loss of a CR16 quality auto-tiler is 3.7% where the highest accuracy loss was at the 4th layer as well. We also calculated the Kbpi required to achieve 0.6 mAP by using linear interpolation. We were able to see that using an auto-tiler reduced the Kbpi requirement by 62.9% to 81.2% on average on CR4 mode, and 81.9% and 92.6% on average on CR16 mode. Taking everything into account, we were able to achieve better and more efficient compression with an auto-tiler when compared to video encoding methods at the same compression ratio.

#### 4.3.3. Effective Compression Ratio Comparison

In order to analyze the compression performance of an auto-tiler, we compared the effective compression ratio of an auto-tiler with existing methods. We defined the effective compression ratio as the range of compression ratios from the highest achievable mAP of that method to the 85% of the maximum mAP of the network. Since we achieved a maximum mAP of 0.72 on the YOLOv3 network that we used on VOC dataset, the effective compression ratio would be the range of achievable compression ratio within the mAP range of 0.72 to 0.612. Note that the maximum mAP will not always be the same across all methods since auto-tiler already compresses the network, and is a lossy compression method. The result is shown in Figure 13 where auto-tiler CR4 and auto-tiler CR16 represents the range of effective compression ratio achieved with fixed CR modes. Auto-tiler CR4-16 on the other hand is not limited to single CR mode and the compression ratio depends on the inference depth. We can observe that the average effective CR range for FPC and tiling method is 1.2×–19.6× and 1.8×–25.5× respectively, while auto-tiler CR4–16 supports an effective CR range of 9.9×–173.3×. From this result, it is clear that auto-tiler supports a much wider and higher range of compression ratio, with its average maximum compression ratio being 8.8× to 6.8× higher than the existing FPC and tiling methods.

#### 4.3.4. Latency Comparison

One of the most important benefits of using an auto-tiler over conventional methods is its higher throughput. While the additional neural network adds a small amount of latency, preprocessing and video coding latency is significantly reduced since the network reduces the dimension of the feature maps. This effect is shown in Figure 14, where we compared the per-image latencies of each step in the overall encoding and inference process.

Each latency component is explained as follows—YOLO inference latency (Pt. 1) is the time it takes to propagate through the network until the partitioned layer for Tiling and FPC method, and output of the encoding part of an auto-tiler for auto-tiling method. Tiling and normalization latency is a latency needed to tile the feature space if needed, and normalize the frame into a format that is supported by video encoders. Video encoding latency and video decoding latency is the time required to encode and decode the processed frames using HEVC video codec. De-tiling and denormalization latency is the time required to de-tile the frame if needed, and denormalize the frames to recover the feature space sent from the partitioned layer. YOLO inference latency (Pt. 2) is the time it takes to propagate through the rest of the partitioned network, including the decoding part of an auto-tiler when measuring the latency for an auto-tiling method. Network latency is not measured and is considered to be the same since we compressed the intermediate activations at the same bitrate of 60 Kbpi during measurement.

For the Tiling and FPC method, the total latency is dominated by tiling and normalization latency. However, when utilizing an auto-tiler, we can see a considerable improvement in that latency. The reason for this improvement is that the latency for tiling or normalization depends on the size of a frame. The frame size of the Tiling method ranges from 832 × 832 × 1 to 1664 × 1664 × 1 and that of the FPC method ranges from 52 × 52 × 256 to 208 × 208 × 64. Auto-tiler on the other hand reduces the dimension to 208 × 208 × 1 or 416 × 416 × 1 depending on the inference depth. This reduced frame size alongside with the removal of tiling process altogether significantly reduces the latency required to tile, normalize, and recover the feature space. Consequently, we can observe an average of 73.2% and 81.3% reduction for CR4 and CR16 mode respectively in terms of overall latency compared to the existing methods.

## 5. Conclusions

In conclusion, we were able to compress the intermediate feature space of deep neural networks more effectively than existing methods by using an auto-tiler as a preprocessor to video encoders. Additionally, by using an auto-tiler we were able to use the bottleneck layer activation by itself without any tiling or de-tiling process during video encoding. This removal of tiling processes allowed us to significantly reduce the total inference latency. Auto-tiler also reduced the blockiness and discontinuity that may be introduced during the existing tiling process, thus reducing accuracy decay at lower bitrates. Furthermore, by supporting variable input and output dimensions we were able to significantly reduce storage space overhead with minimal accuracy loss. Finally, the utilization of an auto-tiler considerably improved the mAP and bitrate overhead during the compression of the intermediate feature space and effectively allowed a much wider range of compression ratio.

In this paper, we illustrated the advantages of AI-based compression in deep intermediate feature space. Our proposed approach, auto-tiler, will allow IoT edge devices in a collaborative intelligence environment to operate more effectively by improving compression efficiency and latency. We however believe that auto-tiler can be further improved on certain points. For one, future works may improve the input and output variability of auto-tiler. Although the proposed auto-tiler in this paper does support multiple input and output dimensions, it is still required to analyze the available dimensions before deployment. Future research may design an omni-dimensional auto-tiler that may support as many dimensions as the conventional codecs. This will allow it to be deployed on any partitioned network without prior dimensional analysis. Another could be to improve the robustness to data losses. Our work is mainly focused on the losses introduced by compression artifacts and assumed no data loss during transmission, as it was beyond the scope of our paper. However, in real-life situations, there may be multiple issues that could result in such losses. Therefore it could be advantageous to consider these issues during the design phase. With these improvements, we speculate that auto-tiler can become an essential technique to be used in IoT devices. To this end, other aspects of auto-tiler must also be explored and need further study.

## Figures and Tables

**Figure 1 sensors-21-00896-f001:**
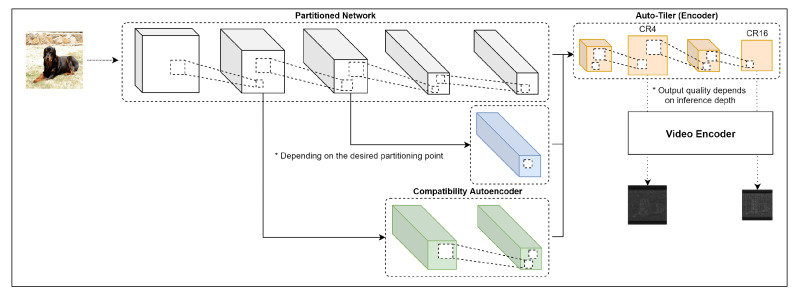
Overview of the proposed auto-tiler encoding process.

**Figure 2 sensors-21-00896-f002:**
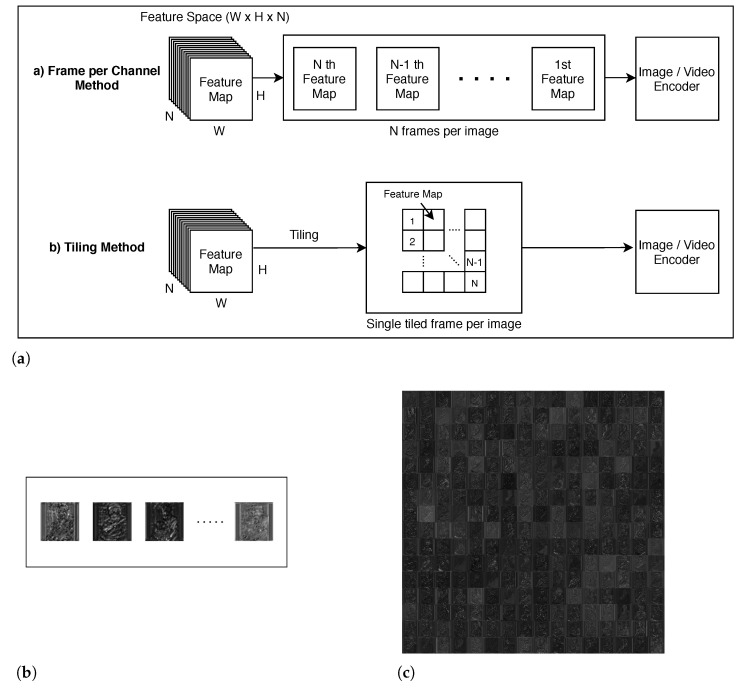
(**a**) Frame per Channel (FPC) and Tiling process, (**b**) visualization of Frame per Channel method, and (**c**) visualization of a tiled feature map.

**Figure 3 sensors-21-00896-f003:**
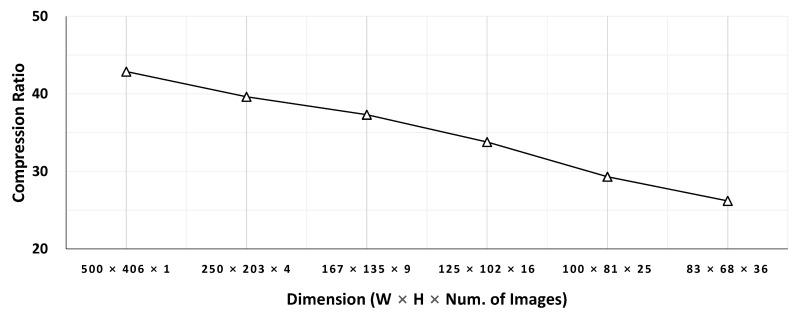
Impact on compression ratio due to splitting.

**Figure 4 sensors-21-00896-f004:**
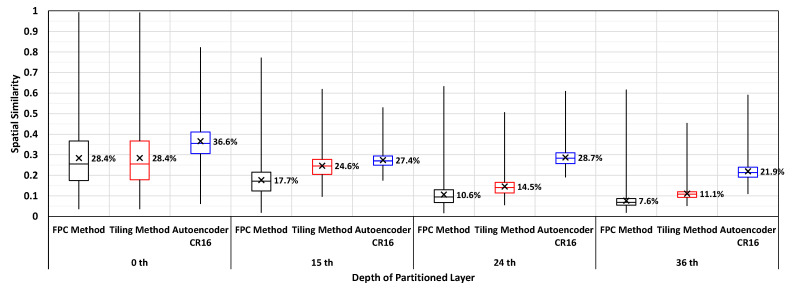
Spatial similarity result of YOLOv3 network at different partitioned layers.

**Figure 5 sensors-21-00896-f005:**
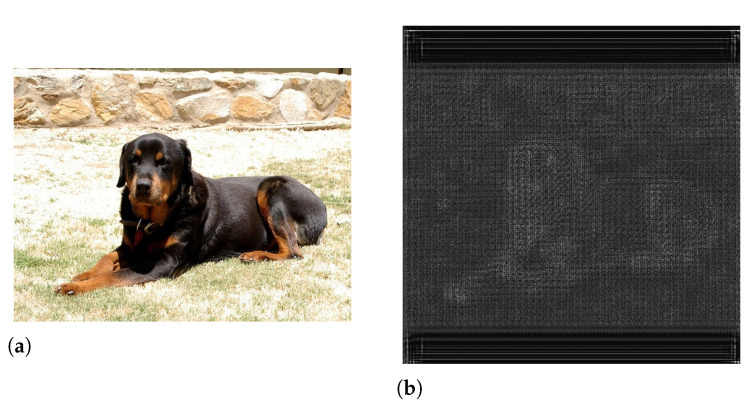
(**a**) Input image and (**b**) a visualized example of an auto-tiled feature map.

**Figure 6 sensors-21-00896-f006:**
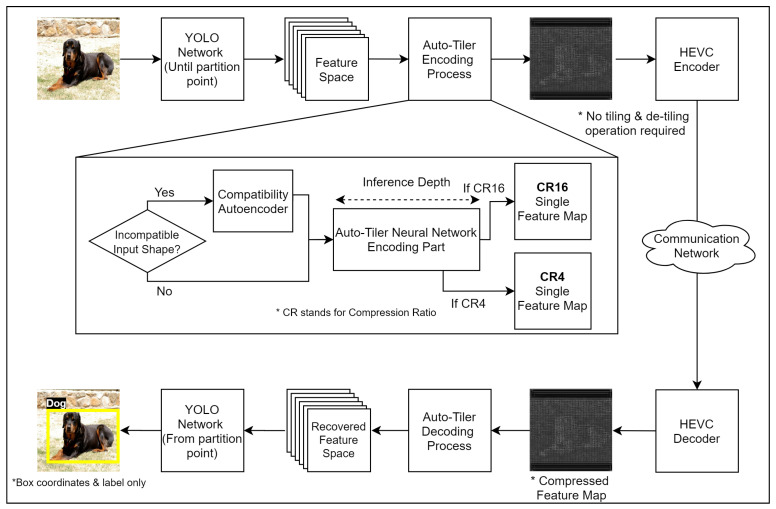
The overview of proposed encoding and decoding process of an auto-tiler.

**Figure 7 sensors-21-00896-f007:**
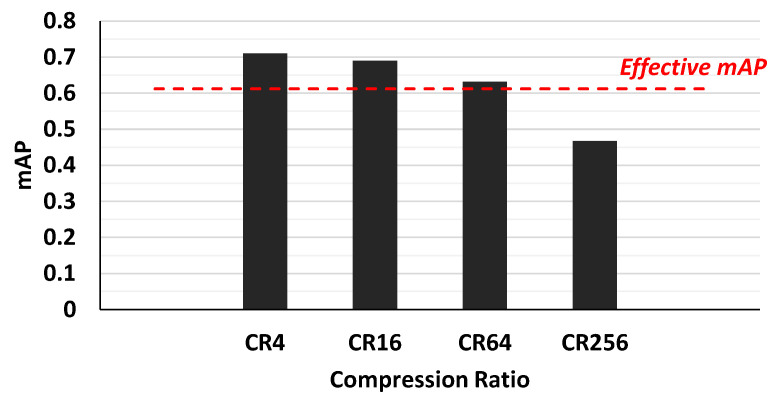
Auto-tiler with compression ratio (CR) of 4–256 and its respective mAPs.

**Figure 8 sensors-21-00896-f008:**
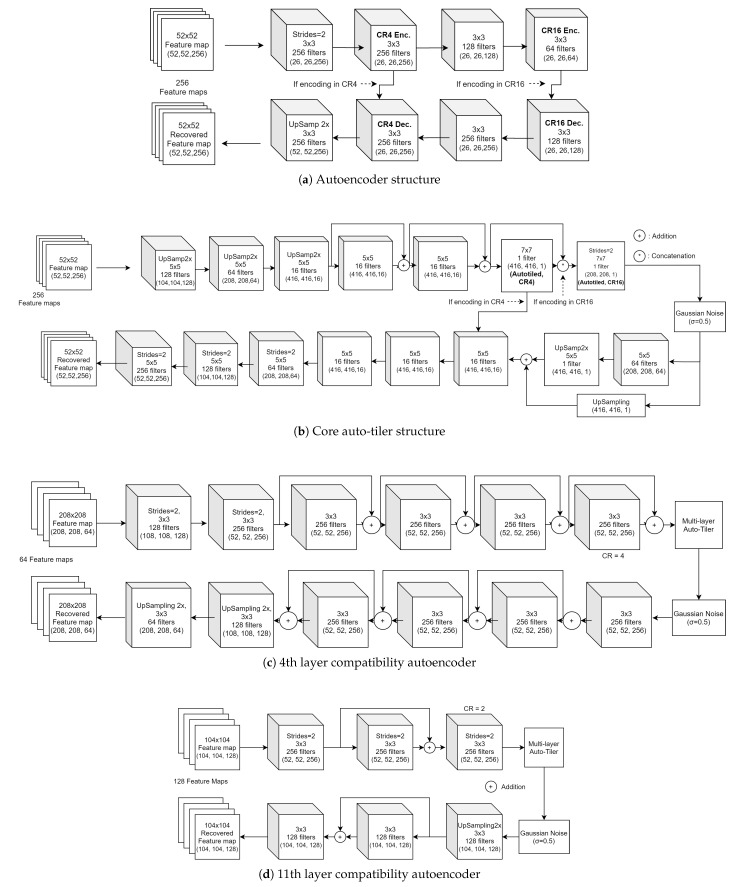
Network structure of (**a**) autoencoder, (**b**) multi-layer auto-tiler, (**c**) 4th layer compatibility autoencoder, and (**d**) 11th layer compatibility autoencoder.

**Figure 9 sensors-21-00896-f009:**
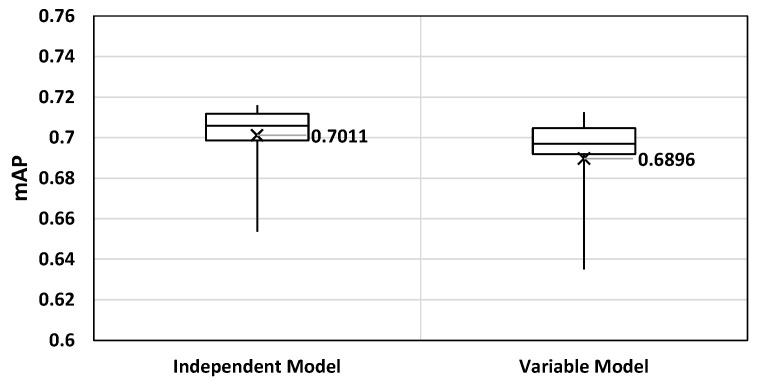
mAP comparison of independent vs variable auto-tiler model.

**Figure 10 sensors-21-00896-f010:**
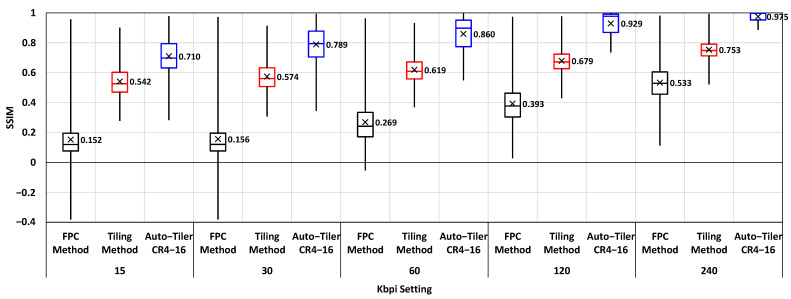
Structural similarity (SSIM) measurement result for FPC, tiling, and auto-tiler method at different Kbpi setting.

**Figure 11 sensors-21-00896-f011:**
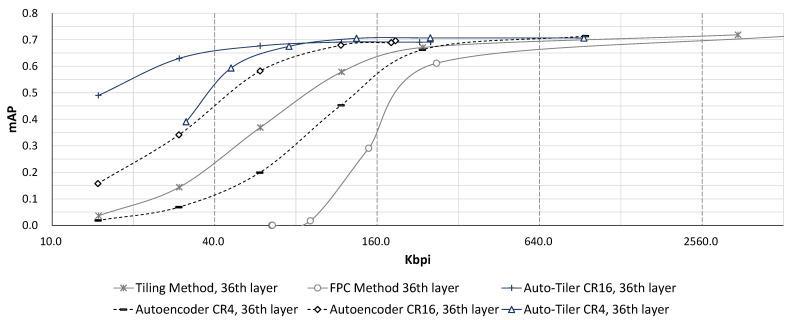
mAP comparison of an auto-tiler, autoencoder, and existing methods at 36th layer.

**Figure 12 sensors-21-00896-f012:**
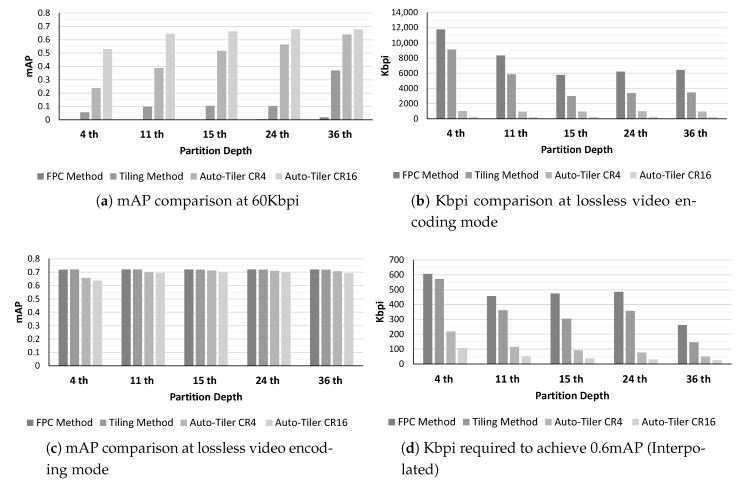
mAP comparison at 60 Kbpi and lossless video encoding mode, Kbpi comparison at lossless video encoding mode, and Kbpi required to achieve 0.6 mAP.

**Figure 13 sensors-21-00896-f013:**
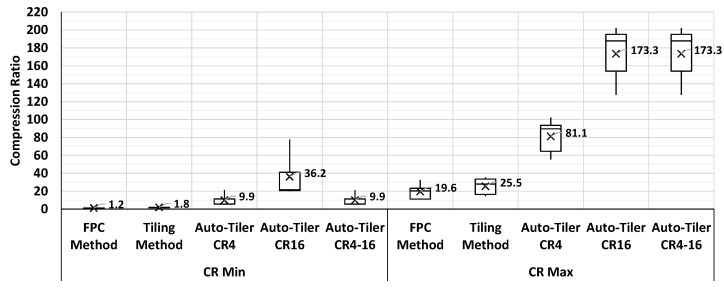
Effective CR comparison of an auto-tiler and existing methods.

**Figure 14 sensors-21-00896-f014:**
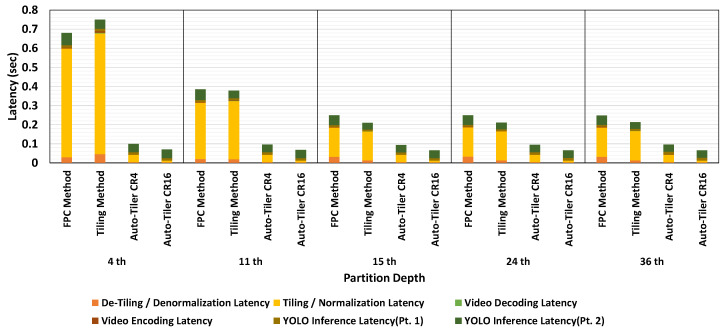
Latency comparison of Tiling, Frame per Channel, and auto-tiling method.

**Table 1 sensors-21-00896-t001:** Storage space overhead comparison of independent vs variable auto-tiler model.

Model	Independent	Variable	Reduction (%)
Encoder	71,002 KB	18,479 KB	73.973%
Decoder	68,592 KB	34,656 KB	49.475%
Total	139,594 KB	53,135 KB	61.936%

## Data Availability

The data analyzed in this study are openly available in the Pascal Visual Object Classes Hompage at http://host.robots.ox.ac.uk/pascal/VOC/index.html, reference number [24,25].

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
