# Peer review of "Auto-Tiler: Variable-Dimension Autoencoder with Tiling for Compressing Intermediate Feature Space of Deep Neural Networks for Internet of Things"

_sensors, 2021, doi:10.3390/s21030896_

Round 1

Reviewer 1 Report

This work introduces an autodecoder method for deep neural network feature space compression. The paper is well-written and the objectives and contributions of this work are clearly presented from the beginning of this paper so the reader can understand.

The results are also quite convincing.

Please check a few typos before the final submission. For example

2.3. Autoencoder as a Preprocessor
Autoencoder is a special type of a neural network which is typically used to reduce the dimensionof the input feature space using a bottleneck layer. They are (Here, you should either say, IT IS, or change the sentence to AUTOENCODERS ARE A SPECIAL...

In my opinion this paper is worth to be published as it is

Author Response

We are deeply grateful to the reviewer for carefully reading our manuscript and for the encouraging comment. It would most definitely help us improve the quality of our manuscript. According to the reviewer’s comment, we revised the paper for any additional typos. We believe that all the typos have been addressed. Please see the details in the attached response.

Reviewer 2 Report

It is very important and good approach to apply new auto-tiler with having preprocessing to have efficient data compress and processing on IOT device and flexibility to customize IOT system.

In your paper, however, I would like to suggest clarifying and show some of evidence to rationalize your paper as following points:

  1. In general, you did show how many data and what data you use your study. It seems that you only apply your method in a specific one test case. If so, it is difficult to rationalize your results at all since it is not new theory proposal.
  2. In Figure 4 and related text, you mention the value of maintaining the spatial similarity in deeper layers in autoencoder. But, for me, it is almost same trend when you compare with tiling and FPC since your results shows spatial similarity from 35-30% to 25-15%. It seems that you need more evidence to justify your explanation.
  3. In figure 5, you should show the value of auto-tier visually. Your example image looks like developing pallets (tiles) with edge detection. Need to have some linkage between Figure 2 and Figure 5.
  4. You need to have some explanation why you only select CR4 and CR16 clearly in your study at Figure 5 (maybe, I miss to read some explanation).
  5. In figure 7, you explain with chapter 3.2.1 and 3.2.2, you explain only 4th and 11th layer. I could not understand why these layers are important to explain in detail.
  6. Figure 8, for me, there is no big difference between independent model and variable since it is only 1.15 % difference. If this number is very big difference, you have to explain more.
  7. In the last text in chapter 4.3.1, you mention about the lossy compression. What is your solution?
  8. Can you please explain more about the relationship between your explanation in your text in 4.3.2 with figure 11? For me, it is difficult to understand the variation of number in figure 11 with last sentence in chapter 4.3.2

Author Response

We are deeply grateful to the reviewer for carefully reading our manuscript and providing critical comments. It would most definitely help us improve the quality of our manuscript. We have made multiple clarification changes and revised the manuscript in accordance with the reviewer's comments. Please see the attached response for further details. We sincerely appreciate the reviewer’s helpful and constructive comments. We believe that the readability and delivery of our manuscript have improved considerably following the revision.

Reviewer 3 Report

I recommend the following adjustments:

  • Figure 4: perform multiple measurements, results in the form of a box-plot.
  • Figure 8: box-plot as in figure 4 (+ average).
  • Figure 11: box-plot as in figure 4 (+ average).
  • It would be appropriate to add:
    • hardware load (see 4.1.1. DNN Model Training; PC AMD Ryzen).

Very nice article!

Author Response

We are deeply grateful to the reviewer for carefully reading our manuscript and for the encouraging comment. It would most definitely help us improve the quality of our manuscript. According to the reviewer’s comments, we changed the mentioned figures into a box-plot format. Additional measurements that we performed is also illustrated in that way if applicable. In addition, we added the hardware compute capabilities (FP32 performances for CPU and GPU) in accordance with the reviewer’s comment. Please see the details in the attached response. We sincerely thank the reviewer for helping us improve the clarity of our manuscript.

Round 2

Reviewer 2 Report

Thank you for your reply with revision of your paper. But, I have still several comments as follow to refine your paper.

1) In your reply in NO 3, I understand your reply in your explanation but for me, it is still difficult to understand in your reserach paper itself. Can you add some of text to explain it using your reply for NO 3. (about figure 5)

2) Is it difficult to add the figure in your reply NO 4 in your paper?

Author Response

We deeply appreciate the reviewer for carefully reviewing our manuscript and providing valuable comments. It has most certainly helped us improve the quality and delivery of our manuscript. Please see the attached response for the details of our revision. We would like to express our sincere gratitude to the reviewer for helping us improve the quality and clarity of our manuscript. We are certain that the reviewer’s comments have helped us greatly improve the readability and delivery of our manuscript.
